# Simulation-Based Comparison of PID with Sliding Mode Controller for Matrix-Converter-Based Dynamic Voltage Restorer under Variation of System Parameters to Alleviate the Voltage Sag in Distribution System

**Abdul Hameed Soomro** [1,*] **, Abdul Sattar Larik** [2]**, Mukhtiar Ahmed Mahar** [2] **and Anwar Ali Sahito** [2]

[1]  Department of Electrical Engineering, Nawabshah Campus, Quaid-e-Awam University of Engineering Science & Technology, Larkano 77150, Pakistan

[2]  Department of Electrical Engineering, Mehran University of Engineering & Technology, Jamshoro 76090, Pakistan

*  Correspondence: abdul.hameed@quest.edu.pk

**Abstract:** A constant power supply is a basic need for each consumer due to the increase in sensitive equipment day by day. As per IEEE standards, a 10% reduction in voltage from the supply voltage is not acceptable and may cause the failure of equipment. Previously, different techniques have been used to alleviate the voltage sag, such as STATCOM, DSTATCOM, SVC, and shunt capacitors, but these devices are connected in parallel, which compensates for the low value of voltage sag, and they have high maintenance costs involved. Compensation for the low and high values of voltage sag is possible through a series-connected device such as a dynamic voltage restorer. In this paper, a matrix converter is presented for DVR to convert AC to AC voltage directly and free from batteries, capacitors, and multiple conversions as needed in a voltage source inverter, resulting in a reduced cost of DVR topology. The DVR is meaningless in the absence of a controller, so it is necessary to select a suitable controller for the satisfactory operation of the DVR under a variation of system parameters. In this paper, the performance of a linear PID controller is analyzed and compared with a nonlinear controller, such as a sliding-mode controller, under variation of power system parameters inorder to select a robust controller that performs satisfactorily for DVR. Earlier trial-and-error methods were used to obtain the parameters of PID gains, but they require a large time to obtain the parameters of the PID gains and there is a chance of inaccuracy. A genetic algorithm was used to obtain the gain parameters, but it has more convergence time and the particle swarm optimization technique has involves less reliability. In this research paper, the sliding surface coefficient parameters such as and Ki for the PI sliding surface of SMC and PID gains are taken through an ant colony algorithm to obtain the robustness of the controllers. The purpose of this paper is to introduce the best DVR topology with reduced cost. MATLAB simulation software was utilized to analyze the performance of the DVR with PID and SMC controllers under different fault conditions and also the THD% of proposed controllers was analyzed through FFT.

**Keywords:** ant colony algorithm; matrix converter; DVR (dynamic voltage restorer); PID and SMC controller; power quality





## 1. Introduction

A constant power supply is a basic need for each consumer due to the increase in sensitive equipment day by day [1]. As per IEEE standards, a reduction in voltage from 10%of the supply voltage is not acceptable and may cause the failure of equipment and revenue loss [2]. The researchers present the different types of FACT devices such as SVC, STATCOM, and shunt capacitors for voltage sag alleviation [3], but they are shunt-connected devices, involve high maintenance cost, and also alleviate the low value of the

voltage sag, which are the drawbacks of these devices [4]. Active and reactive powers are needed for the alleviation of the low and high value of sag voltage [5], which is only possible through the application of a series-connected device [1]. The dynamic voltage restorer (DVR) is connected in series with the load in order to compensate the voltage sags of small and large value [6]. The researchers present DVR topologies supported with batteries and capacitors for the supply of DC power to the inverter circuit of the DVR [7], but the drawbacks of these topologies is the high cost of maintenance and replacement [8]. In battery-supported DVRs, the multiple conversion of power supply is needed, which increases the complexity of the system [9]. In this paper, a matrix converter is presented which is free from batteries, capacitors, and multiple conversions [10]. It converts AC to AC voltage directly [11], which results in the low cost and less complexity of the DVR topology [12]. The DVR is meaningless in the absence of a controller [4], so it is necessary to select a suitable controller for the satisfactory operation of the DVR under normal conditions as well as for variation of system parameters [7]. In the past, various DVR topologies were presented, such as PID controller with voltage source inverter [13], PID controller with matrix converter [14], and PID controller with PV based voltage source inverter [15],but the control approach of a PID controller is not fit for DVR topologies under variation of power system parameters because the PID controller is a linear controller [14] and the selection of PID controller under fault conditions [14] for DVR topology is questionable. Earlier, a sliding mode controller with a voltage source inverter [7] was presented, but the drawback of such a topology is that it involves high maintenance and replacement cost for the DVR topology due to the utilization of an energy storage device for an inverter circuit [8] to convert DC into AC, and involving multiple conversions [5]. A sliding mode controller with AC chopper [16] was presented to control the dynamics of the DVR, but the drawback is that an AC chopper cannot control the output frequency as a matrix converter [17]. DVR topology with a matrix converter and the control strategy of a sliding mode controller has not been presented [10]. The performance of a DVR topology with a matrix converter and the control strategy of a PID controller is analyzed and compared with a nonlinear controller such as a sliding mode controller under system parameter variations under fault conditions. Different types of techniques were used by researchers [18,19] to obtain the values of the PID gains, and the traditional PID controller is mainly tuned with the Ziegler–Nichols method [20]. In this method, more time is spent to obtain the values of the PID gains and sometimes the generated values give results with large overshoot [21]; therefore, retuning is necessary before applying the control to the device [22]. Pole placement and Cohen–Coon methods can be utilized to obtain the values of PID gains, but they depend on low-order model estimations [23], if a real-coded genetic algorithm is used the convergence time is greater [24], and the PSO technique is less reliable [25,26]. It is necessary to control the operation of DVR to select the optimum values of the sliding surface coefficient parameters such as $K_p$ and $K_i$ for the PI sliding surface of SMC [27] and the tuning parameters for the PID controller. An iterative method, such as the Ant colony algorithm [28], is the best choice [29] in order to save time, reduce insecurity, and provide more reliability [29]. In this paper, the parameters of a PI sliding surface for SMC and PID gains are taken through an ant colony algorithm [30] to obtain the robustness of the controller. The purpose of this paper is to introduce the DVR topology of less cost with a robust controller in order to perform satisfactorily under variation of system parameters. MATLAB simulation software is utilized to analyze the performance of the DVR with PID and SMC controllers under different fault conditions and the THD% of proposed controllers is analyzed through FFT.

## 2. Dynamic Voltage Restorer

The dynamic voltage restorer is connected in series with the sensitive load in order to inject the required voltage in case of voltage sag [1]. The matrix converter, LC filter, voltage injection transformer, conventional PID controller, and a sliding mode controller are the components of the proposed DVR as shown in Figure 1. The matrix converter converts AC to AC voltage directly [10]. In this paper a three-phase matrix with space

vector modulation technique is proposed to control the switching sequence of the matrix converter [31]. A distorted waveform is achieved in the output of matrix converter [32] because semiconductor switches are available in the matrix converter [33]. The problem of distorted waveforms can be avoided by the addition of a filter circuit [34]. The main cause of the addition of the voltage injection transformer is to inject the necessary voltage to the sensitive load in the case of sag voltage [4,12]. The PID and SMC controllers are used to control the dynamics of DVR under nonlinearities.

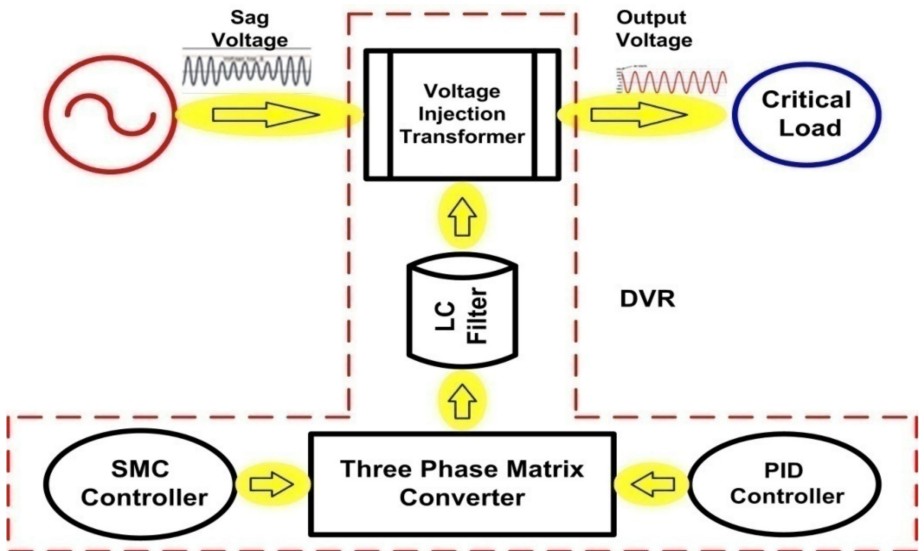

**Figure 1.** Proposed DVR.

### 2.1. Matrix Converter

In this research paper, a three-phase matrix converter device is utilized. It consists of nine bidirectional IGBT switches that are connected in such a way that they form a matrix [17]. It has the capability to convert AC to AC voltage directly and free from an energy storage device, charge up circuit, and multiple conversions [11], which results in reduced cost and less maintenance. Figure 2 shows the direct three-phase matrix switches with common source configurations; in each phase, three switches are connected. The load is connected to check the performance of the converter. Bidirectional switches must be connected in such a way that the input voltage and current are connected with the output voltage and current for the proper commutation sequence [35]. A matrix converter has high-quality input current signals, controllable input power factor, and compact design. DVR topology utilizes the direct matrix converter and obtains the compensation energy from the same power supply for the balanced and unbalanced disturbances [36].

### 2.2. Park Transformation

Direct-quadrature-zero ($dq_o$) transformation is also called Park transformation, which is presented mathematically for the simplicity of controller design and easy analysis of the electrical engineering problems concerned with the three-phase system [37]. In this technique, electrical quantities of three phases are reduced to two-phase electrical quantities. As shown in Figure 3, in the first step, three-phase voltages/currents are given to the $abc/\alpha\beta$ block and are in line with the phase lock loop and converted to $\alpha\beta/dq$. The reference and actual values are added and difference is sent to the control circuit, after which the $dq$ components are converted to $\alpha\beta$, and finally $\alpha\beta$ is converted into abc and the PWM signal is sent to the converter for the operation of the switches [4].

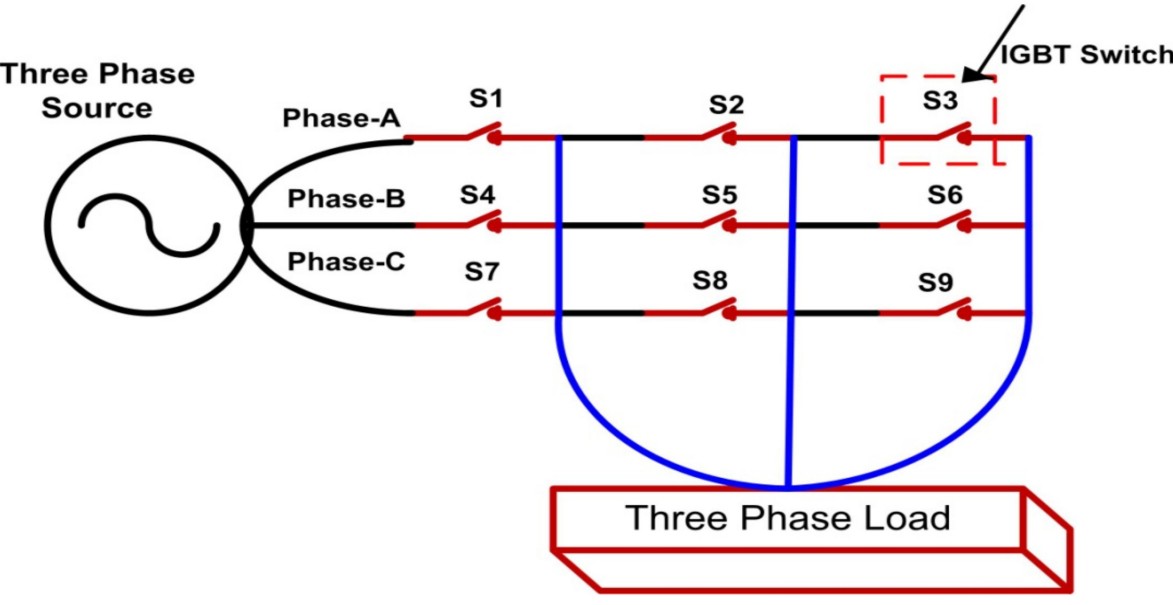

**Figure 2.** Direct three-phase matrix converter.

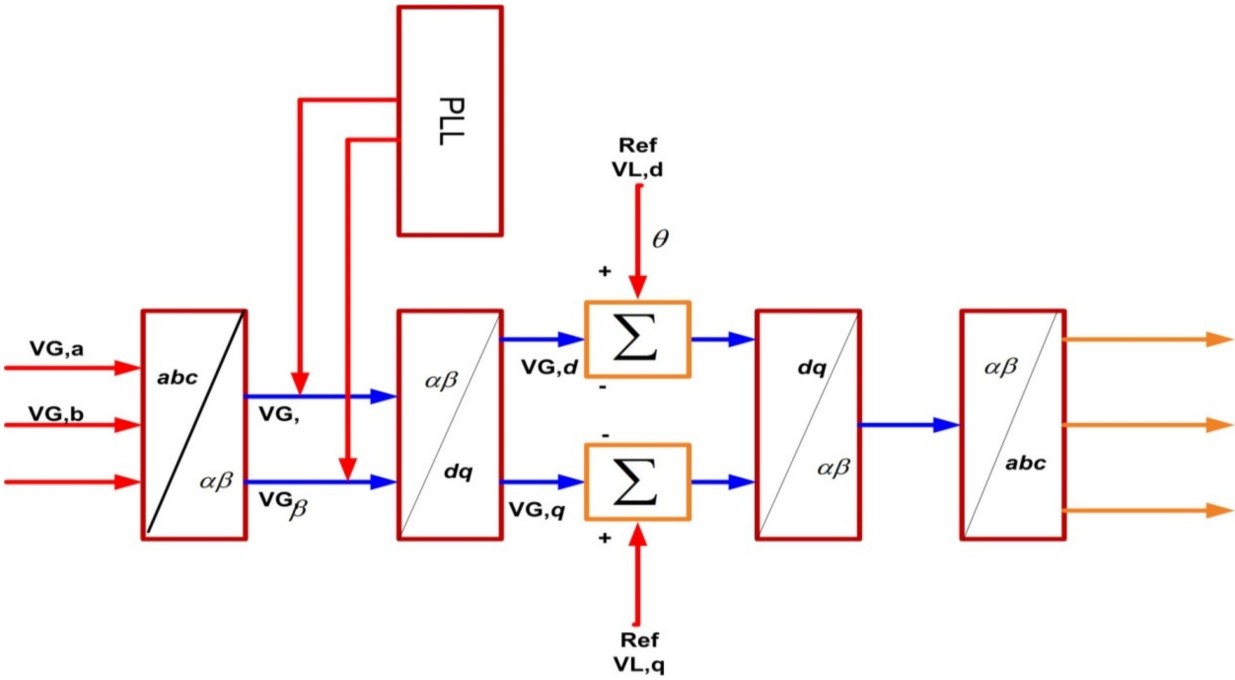

**Figure 3.** Coordinated dq$_o$ transformation.

### 2.3. Voltage Injection Technique

The voltage injection technique is necessary for DVR topology because the value of voltage and phase angle change after the occurrence of voltage sag [5]. A pre-sag voltage injection technique is shown in Figure 4 and proposed in this paper because the phase angle and magnitude remain the same in this technique and the load is free from voltage disturbance [4].

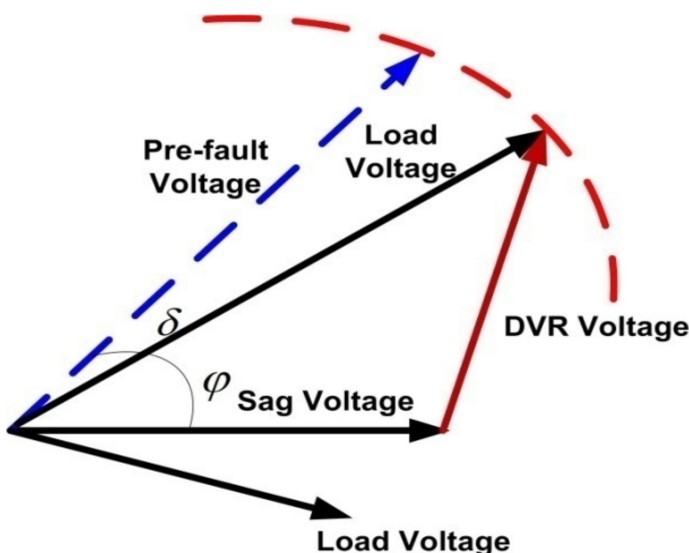

**Figure 4.** Proposed voltage injection technique.

In the pre-sag voltage injection technique, when voltage sag is detected, the DVR will come into its active mode and the needed voltage will be provided to the load [4]. In the case of voltage sag, the phase angle shift δ = 0 and

$$V_{INJ} = V_{Presag} - V_{Postsag} \tag{1}$$

where

$V_{INJ}$ is the injection voltage.
$V_{Presag}$ is the pre-sag voltage.
$V_s$ is the supply voltage.
$V_L$ is the load voltage.

The needed power is calculated through the Equation (2):

$$P_{INJ} = \sqrt{3} * V_{INJ} * I_L + \cos(\theta_L + \theta_{INJ}) \tag{2}$$

## 3. Control Circuit for DVR Topology

The control circuit is very essential for the topology of DVR; without the control circuit, the topology of DVR is meaningless [4,17], so proper selection of a controller is necessary for the operation of DVR under balanced and unbalanced conditions. The PID controller is mostly preferred for steady state conditions [21,38] but fails in the case of fault conditions [4]. In a PID controller, the proportional gain helps to reduce the error response, integral gain eliminates the steady state error, and the derivative gain improves the stability of the system [39]. The performance indexes of the PID controller depend on the four parameters such as overshoot, settling time, rise time, and integer absolute error. PI and PID controllers have been used for the topology of the DVR, but they have well known limitations under variations of system parameters and take a long time in settling [4,7,37,39,40]. A sliding mode controller is a non-linear controller with more advantages such as stability under parameter variations, robustness, good dynamic response, and ease of implementation [7,16]. SMC design depends on the switching function and control law [41]. A fast switching response is essential to move the system variables to zero.

The equation for a sliding mode controller is given in Equation (3).

$$u = -sign(S) \tag{3}$$

$$S = \dot{X} + a.|X| \tag{4}$$

where $S = 0$ and $a$ is the variable

$$\dot{X} = -a.X \tag{5}$$

The sliding phase/surface in which the system slides and makes the system variable to reach at zero $S < 0$. The reachability phase is simple, as it says that any controller that satisfies the condition $S\dot{S} < 0$ will reach the sliding surface and be known as the reachability phase, but due to the inertia of the system, the reachable phase is not equal to zero $S > 0$ and chatter with back-and-forth motion along the sliding surface, which is known as chattering or oscillations, is shown in Figure 5. The problem of chattering can be avoided by using a hysteresis band [7]. The $\dot{S}S$ are opposite to each other with $S = 1$ and $\dot{S} = -1$ and $\delta(t)$ is the switching function of the matrix converter, either 1 or $-1$, as given in Equation (6).

$$\delta|t| = \left\{ \begin{array}{l} 1(ONtime)\ for\ S < 0 \\ -1(OFFtime)\ for\ S < 0 \end{array} \right\} \tag{6}$$

when the SMC slides and the system variable reaches zero, then the system will be stable [42].

$$u_f = -V_d. - sign(S) \tag{7}$$

where $u_f$ is the control input to the matrix converter and $V_d$ is the DVR voltage.

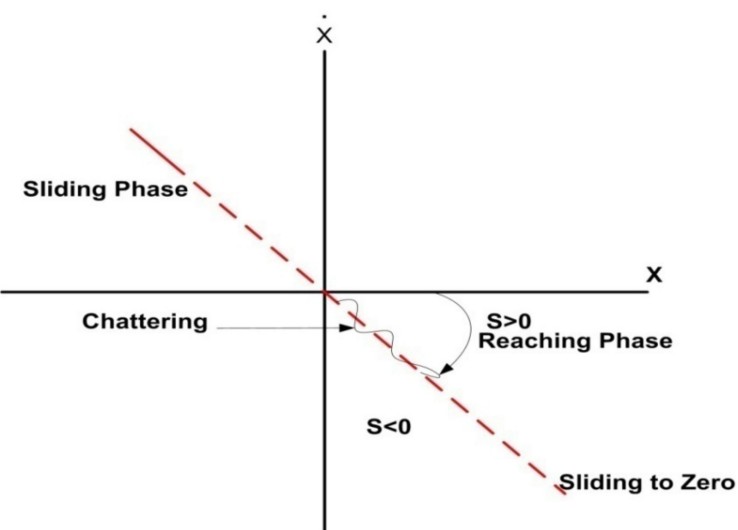

**Figure 5.** Phase plot of SMC.

The sliding surface is calculated through Equation (8),

$$S_{(t)} = K_p e + K_i \frac{di}{dt} e \tag{8}$$

where $e$ is the error, and $K_p$, $K_i$ are the PI gains.

State space formulation was utilized in the sliding mode, but now a day's hysteresis relay is commonly used for the sliding mode because this technique is simple and easy to implement [7]. Therefore, a hysteresis relay sliding mode is selected in this research paper.

## 4. Ant Colony Algorithm

An ant colony algorithm is an advanced empirical bionic algorithm [42]. In this algorithm, the ants are moving from the nest to the food center and select the smallest path through the sensing of pheromones [30]. As shown in Figure 6a, the ants meet at a point and have no idea how to select the shortest way to reach the food center [43]. There are two tracks, one on the upper side and the other on the lower side, so half the ants move to the upper track and half the ants move to the lower track [42]. As shown in Figure 6b,c,

assuming that the speed of ants moving is the same, the distance upto the food center is smaller through the upper track and greater through the lower track, and ants select the regular path to accumulate pheromones faster [19]. After a short period of time, it has seen that large quantities of pheromones are accumulated through the upper track and a smaller quantity of pheromones are accumulated through the lower track by the ants [30], as shown in Figure 6d. The large quantity of pheromones help the new ants that want to move towards the food center [29]; in this way all ants will move through the shorter track [42].

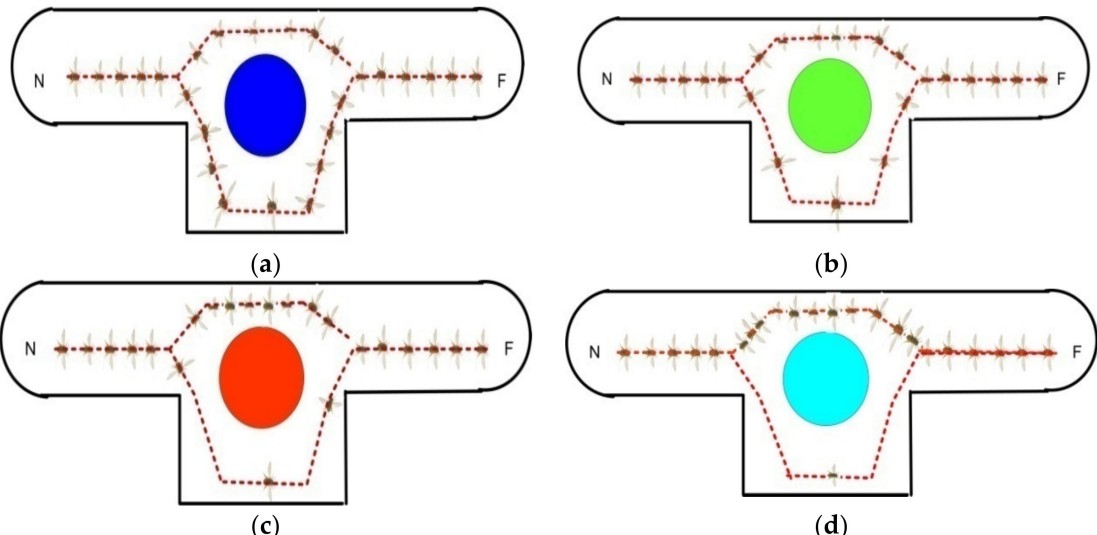

**Figure 6.** (**a**–**d**) Movements of ants from nest to food center.

### 4.1. Cost Function

The following three cost functions are suggested for the tuning of the PID controller [44]. The first cost function is given in Equation (9), and through this we can reduce the integrated square error.

$$f_1 = \int_0^\infty (e^t dt)2 \tag{9}$$

The second cost function is given in Equation (10). In this cost function the rise time, maximum overshoot, steady state error and the settling time are summed up and the sum of these will be utilized to assess the second cost function.

$$f_2 = \frac{1}{\left[ C_1 (t_r - t_{rd}) + C_2 \left( M_p - M_{pd} \right) + C_3 (t_s - t_{ds}) + C_4 (e_{ss} - e_{ssd}) \right]} \tag{10}$$

The third cost function is given in Equation (11). The design of the PID controller depends on this cost function [42]. The rise time, settling time, overshoot and steady state mainly depend on the selection of the beta, and the value of $\beta$ is 0.2.

$$f_3 = \frac{1}{\left( 1 - e^{-\beta} \right) \left( M_p + e_{ss} \right) + e^{-\beta}(t_s - t_r)} \tag{11}$$

### 4.2. Parameters for ACO

The PID and SMC controllers are designed with ACO technique to choose the better values of the PID gains and sliding surface coefficient parameters such as $K_p$ and $K_i$ for the PI sliding surface of SMC in order to achieve the robustness of the controller and small transient response [30]. In this paper, 50 ants and 50 iterations with 0.8 and 0.2 values of

Alpha (α) and Beta (β) are selected with 1000 nodes in an ant colony optimization technique. All the computation was carried out through MATLAB Simulink software.

### 4.3. Parameters of Controller Gains

Table 1 shows the parameters of controller gains as taken through the ant colony optimization technique.

**Table 1.** Parameters of controller gains.

| S.No | Parameter | Value |
|------|-----------|-------|
| 1 | Proportional gain | 6.87688 |
| 2 | Integral gain | 0.85085 |
| 3 | Derivative gain | 10 |

## 5. Proposed Power System with Simulation Results

The parameters of Table 2 were utilized in an electrical power distribution system with the proposed DVR, as shown in Figure 7. The performance of DVR under different fault conditions is analyzed with application of the PID and SMC controller through MATLAB Simulation software.

**Table 2.** Power system and DVR parameters.

| S.No | Parameter | Value |
|------|-----------|-------|
| 1 | Source voltage | 11 KV |
| 2 | Step-down transformer | 11 KV/400 V |
| 3 | Resistance of the line | 0.789 |
| 4 | Inductance of the line | $15.48 \times 10^{-6}$ |
| 5 | Load (3-phase) | 5 KW |
| 6 | Filter inductance | 4.5 mH |
| 7 | Filter capacitance | 2 μF |

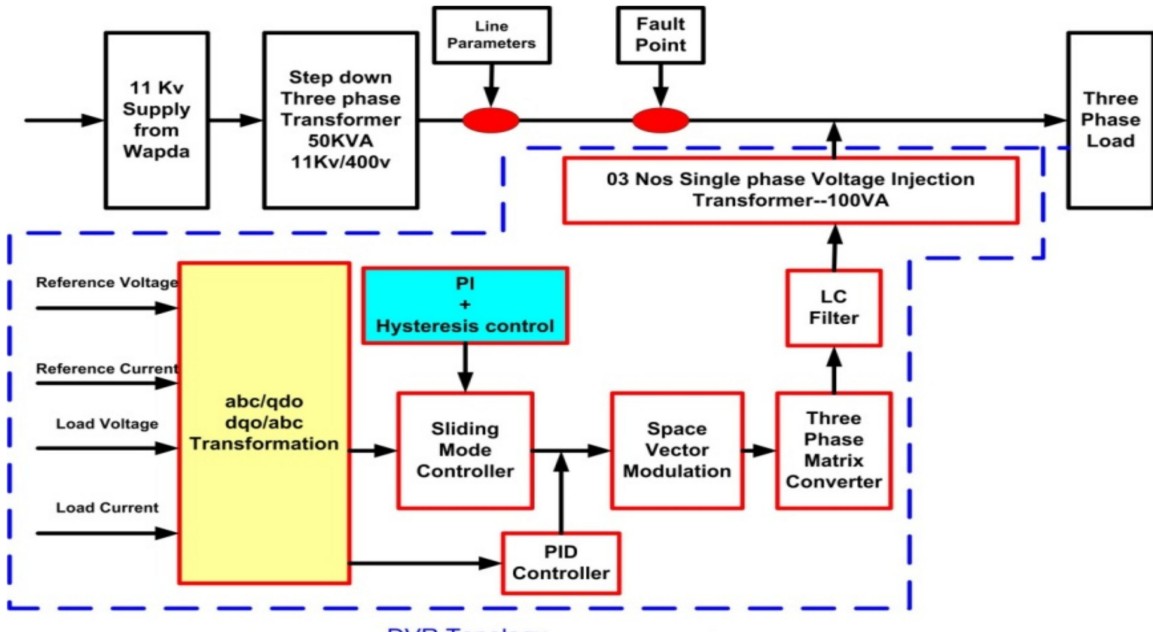

**Figure 7.** Single line diagram of distribution system with proposed DVR.

### 5.1. MATLAB Simulation Results

5.1.1. Case1: Three Phase-to-Ground Fault with PID Controller

In case1, faults occurred in all three phases and voltage was reduced to 70% of the supply voltage. In this case the DVR comes into voltage injection mode and adds the required voltage to the load; the supply voltage is shown in Figure 8a, the load current is shown in Figure 8b, the load voltage is shown in Figure 8c, and the voltage injected by DVR is shown in Figure 8d. It is seen that the output load current and voltage increases beyond normal values and the PID controller settled at time of 0.04 s.

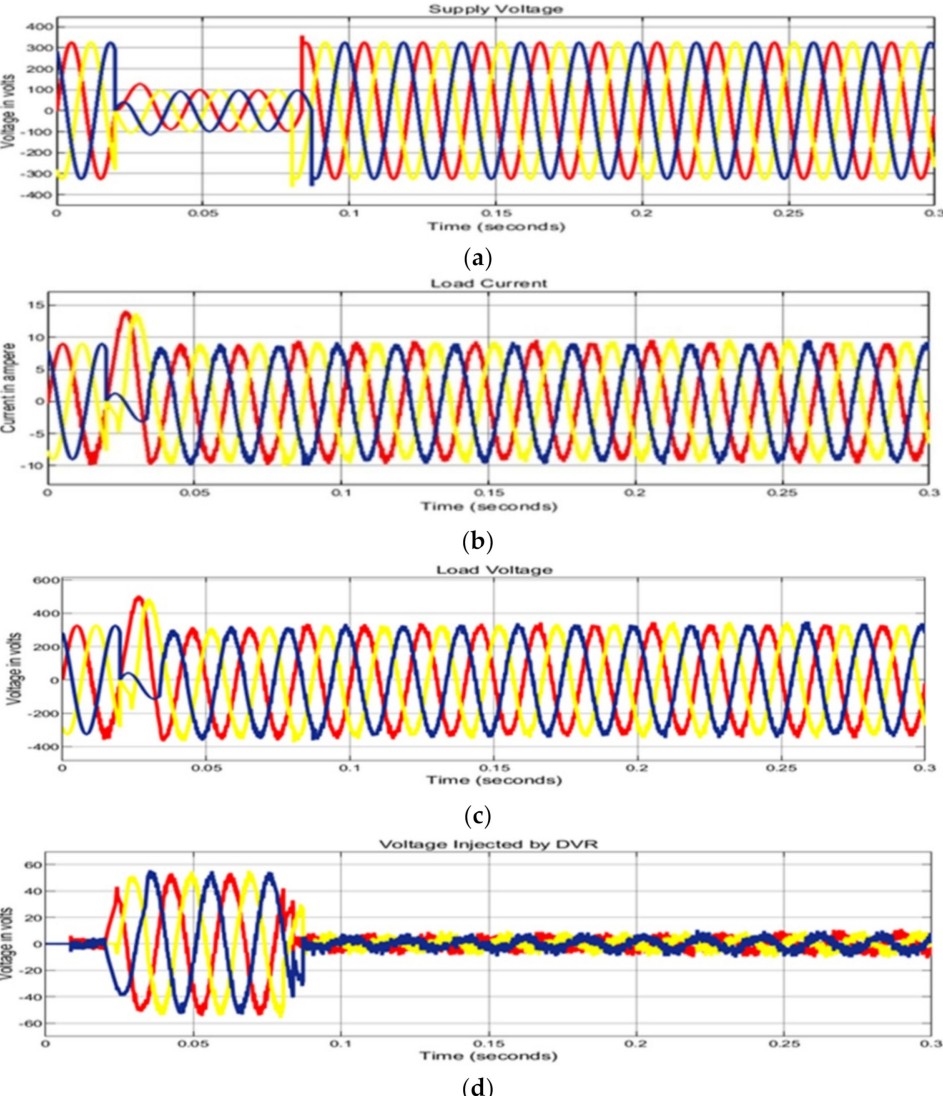

**Figure 8.** (**a**) Supply voltage during three phase-to-ground-faults with PID controller. (**b**) Load current during three phase-to-ground-faults with PID controller. (**c**) Load voltage during three phase-to-ground-faults with PID controller. (**d**) Voltage injected by DVR during three phase-to-ground-faults with PID controller.

5.1.2. Case2: Double Line-to-Ground Faults with PID Controller

In case 2, phases B and C are under fault and phase A is in healthy condition, then the voltage in phases B and C is reduced to 70% of the supply voltage. In this case, the DVR comes into voltage injection mode and adds the necessary voltage to the load; the supply voltage is shown in Figure 9a, the load current is shown in Figure 9b, load voltage is shown in Figure 9c, and the voltage injected by the DVR is shown in Figure 9d. It is seen that

the output load current and voltage increase beyond normal values and the PID controller settled at the time of 0.04 s.

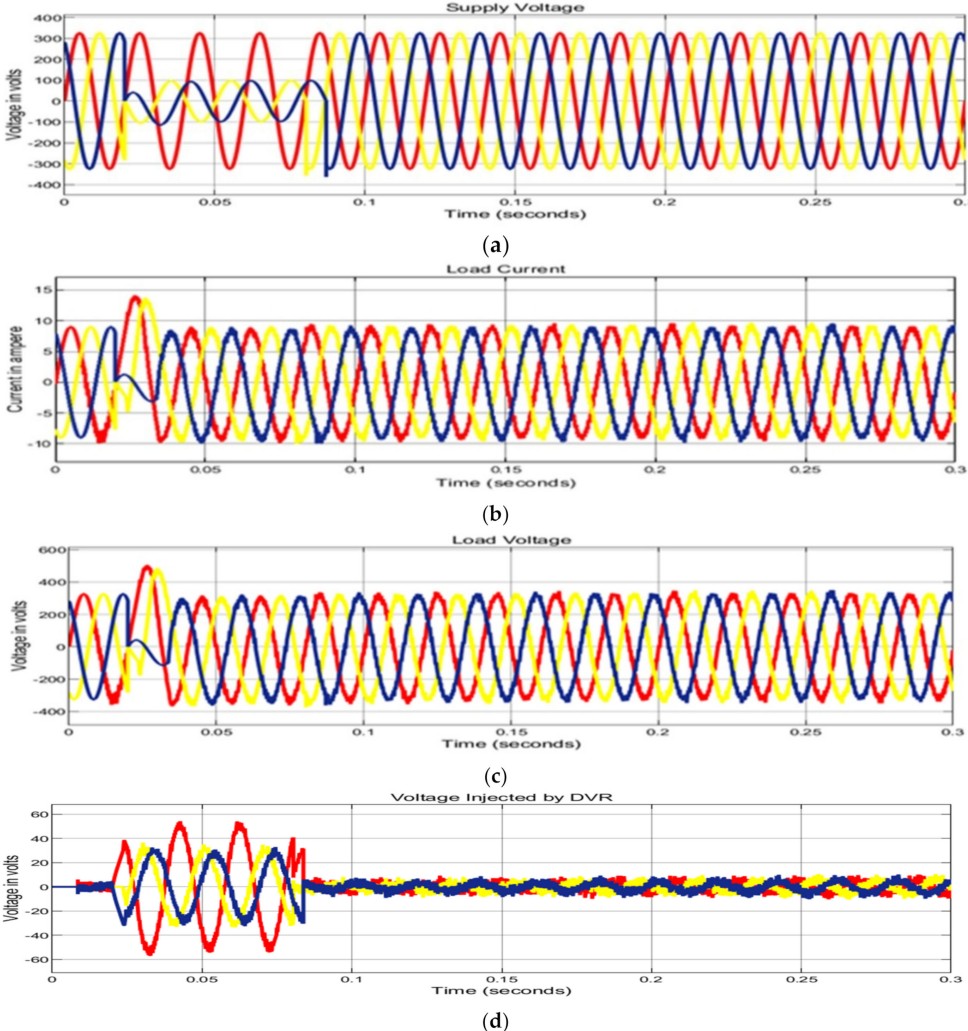

**Figure 9.** (**a**) Supply voltage during double line-to-ground faults with PID controller. (**b**) Load current during double line-to-ground fault with PID controller. (**c**) Load voltage during double line-to-ground fault with PID controller. (**d**) Voltage injected by DVR during double line-to-ground fault with PID controller.

5.1.3. Case3: Three Phase-to-Ground faults with SMC Controller

In case3, faults occur in all three phases and the voltage is reduced to 70% of the supply voltage. In this case the DVR comes into voltage injection mode and adds the required voltage to the load; the supply voltage is shown in Figure 10a, the load current is shown in Figure 10b, the load voltage is shown in Figure 10c, and the voltage injected by the DVR is shown in Figure 10d. It is seen that the output load current and voltage are maintained with normal values and the controller settled at the time from where the fault time started, i.e., 0.02 s.

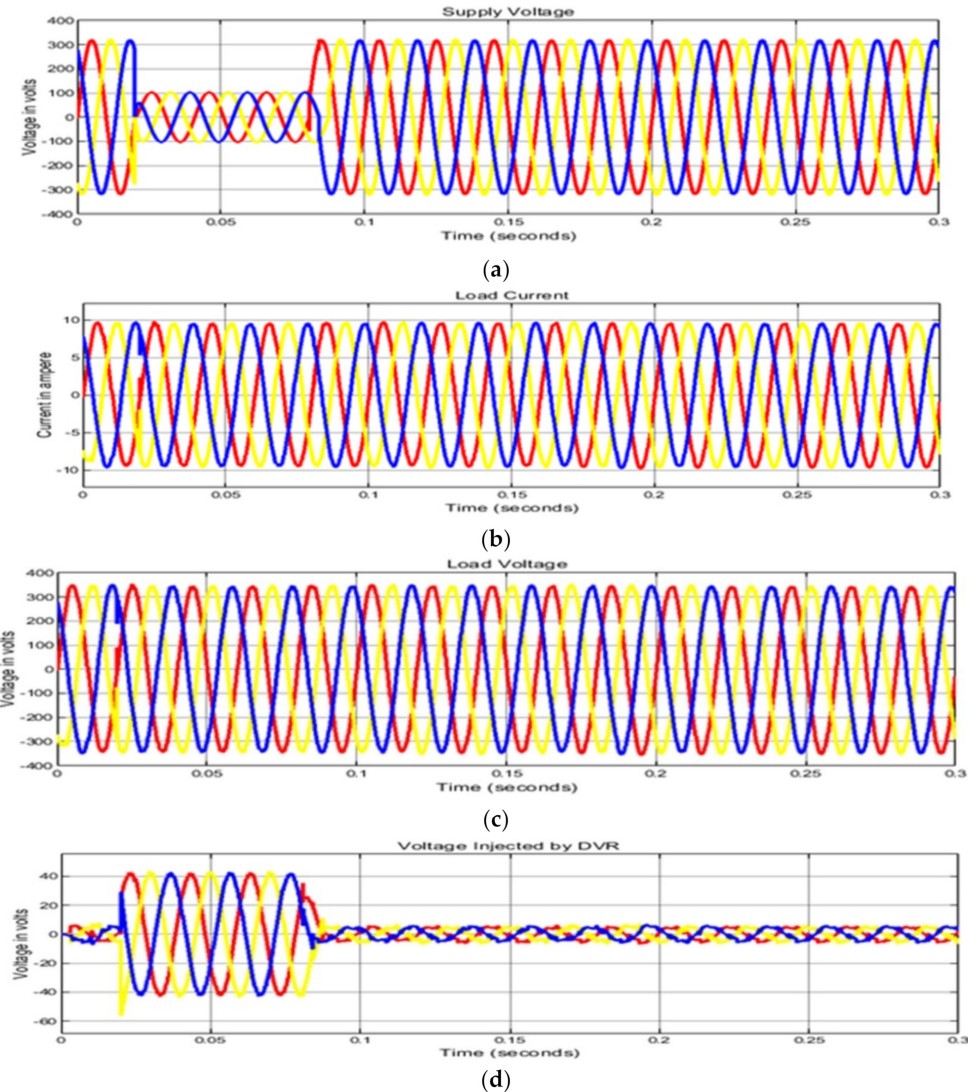

**Figure 10.** (**a**) Supply voltage during three phase-to-ground faults with SMC controller. (**b**) Load current during three phase-to-ground fault with SMC controller. (**c**) Load voltage during three phase-to-ground fault with SMC controller. (**d**) Voltage injected by DVR during three phase-to-ground fault with SMC controller.

### 5.1.4. Case4: Double Line-to -Ground fault with SMC Controller

In case-4, phases B and C are under fault and phase A is in healthy condition, then the voltage in two phases, B and C, is reduced to 70% of the supply voltage. In this case the DVR comes into voltage injection mode and adds the necessary voltage to the load; the supply voltage is shown in Figure 11a, the load current is shown in Figure 11b, the load voltage is shown in Figure 11c, and the voltage injected by the DVR is shown in Figure 11d. It seen that the output load current and voltage are maintained with normal values and the controller settled at the time when the fault started, i.e., 0.02 s.

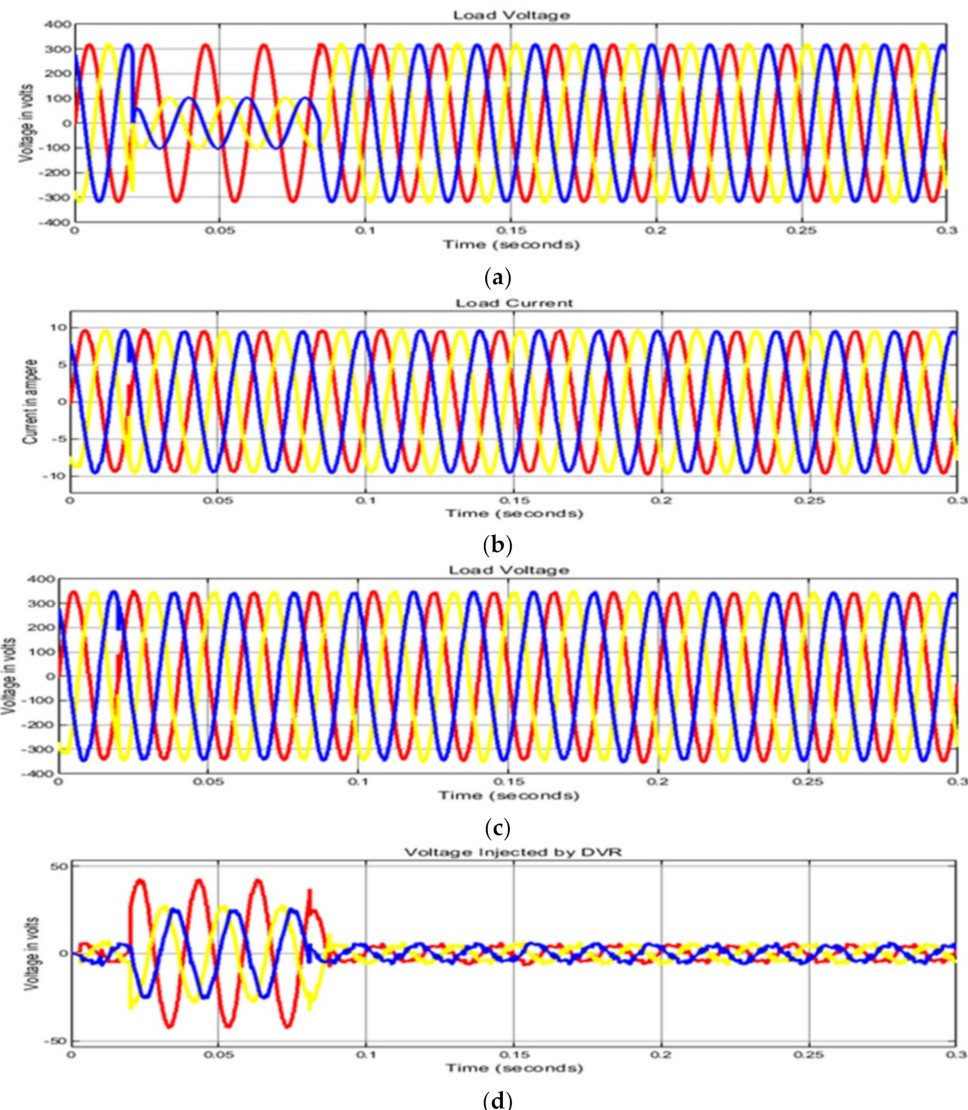

**Figure 11.** (**a**) Supply voltage during double line-to-ground faults with SMC controller. (**b**) Load current during double line-to-ground fault with SMC controller. (**c**) Load voltage during double line-to-ground fault with SMC controller. (**d**) Voltage injected by DVR during double line-to-ground fault with SMC controller.

## 6. Total Harmonic Distortion (THD%)

THD% analysis at the output of the load voltage is a very important factor for the assessment of power quality in the power distribution system. The allowable value of THD% is 5% as per IEEE standard 519-1992 [7]. The THD% of the proposed system was analyzed through FFT. The total harmonic distortion in the output of load voltage without the proposed topology of the DVR under fault conditions is shown in Figure 12a. In this condition, the THD% is 44.28% of the fundamental frequency. The total harmonic distortion in the output of load voltage is analyzed through FFT with the proposed topology of the DVR with a PID controller is shown in Figure 12b, and which is 12.41% of the fundamental frequency. The total harmonic distortion in the output of the load voltage is analyzed through FFT with the proposed topology of DVR with a SMC controller is shown in Figure 12c and which is 1.99% of the fundamental frequency.

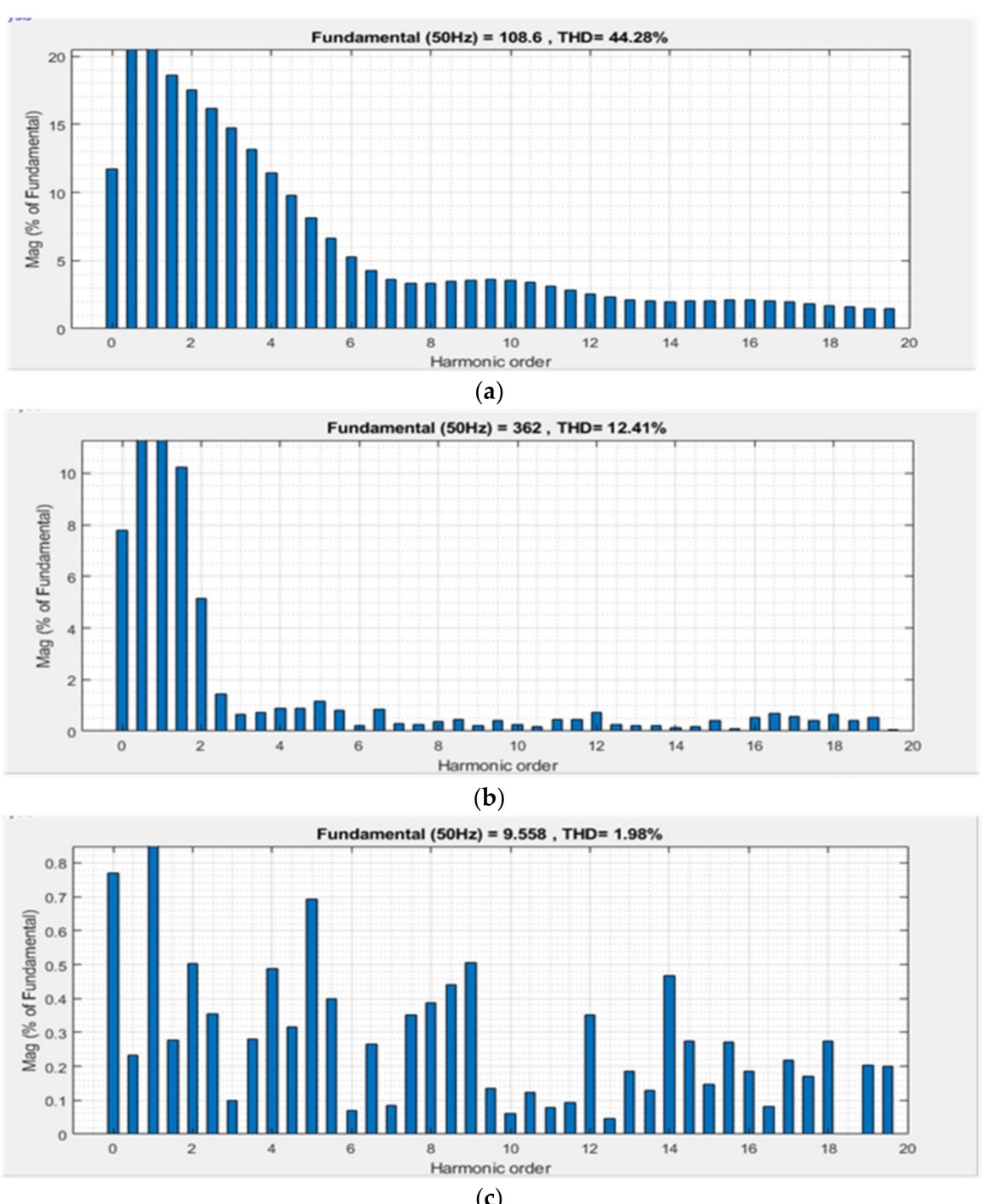

**Figure 12.** (**a**) THD% during fault conditions without DVR. (**b**) THD% with PID controller. (**c**) THD% with SMC controller.

## 7. Discussion

It is seen from the simulation results of DVR topology with a PID controller that the proposed control approach is not fit for the topology of DVR under fault conditions due to large overshoots in the voltage and current and a settling time that starts at 0.04 s, which may result in equipment failure, stoppage of machines, and revenue loss because most of the equipment is very sensitive and cannot accept the small variation in voltage. The THD% of the PID controller with the proposed DVR is 12.41% of the fundamental frequency, which is not acceptable as per IEEE standards. It is seen in the simulation results of DVR topology

with SMC that there is no overshoot in the voltage, and also the THD% is 1.98%, which is within the limit as per IEEE standards. The performance of both controllers is tabulated in Table 3.

**Table 3.** Performance of Controllers with Proposed DVR Topology.

| Parameters | PID Controller | SMC Controller |
|---|---|---|
| Supply voltage | 327v | 327v |
| Sag voltage | 227v | 227v |
| Rise time | 0.025 s | - |
| Settling time | 0.04 s | 0.02 s |
| Overshoot in voltage | 173v | - |
| Overshoot in load current | 5 A | - |
| THD% | 12.41% | 1.98% |

## 8. Conclusions

The DVR topology with direct AC to AC matrix converter and control application of linear controller (PID) and nonlinear controller (SMC) is discussed and the performance of the DVR is analyzed through MATLAB simulation software under different fault conditions, and the THD% is analyzed with FFT. For the robustness of the controllers, the parameters of the PI sliding surface for SMC and PID gains were taken through an ant colony algorithm. It is seen from the simulations and THD% results that the PID controller is not a fit controller for the topology of DVR under system parameters variation due to the large overshoot in current and voltage, as well as the greater time expenditure in settling and the fact that the THD% is not acceptable as per IEEE standards. Therefore, it is seen that the performance of the nonlinear controller (SMC) is satisfactory under parameter variation and the THD% is also within the limit as per IEEE standards. It is concluded that the DVR topology based on a matrix converter and the control approach of a sliding mode controller is the best DVR topology under system parameter variations. This DVR topology has no need for energy storage devices, charge up circuits, and multiple conversions, which results in less cost due to the direct conversion of AC to AC voltage is carried out through the matrix converter.

**Author Contributions:** Conceptualization, Methodology, and Investigation A.H.S.; Data curation, and Formal analysis A.S.L.; Validation, and supervision M.A.M.; Software, and Writing, review & editing A.A.S. All authors have read and agreed to the published version of the manuscript.

**Funding:** This research received no external funding.

**Data Availability Statement:** All data are contained within the article.

**Acknowledgments:** The authors would like to thank QUEST, Nawabshah, Pakistan and MUET, Jamshoro, Pakistan for the research facility provided.

**Conflicts of Interest:** The authors declare no conflict of interest.

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
