# Peer review of "Simulation-Based Comparison of PID with Sliding Mode Controller for Matrix-Converter-Based Dynamic Voltage Restorer under Variation of System Parameters to Alleviate the Voltage Sag in Distribution System"

_sustainability, doi:10.3390/su142114661_

Round 1

Reviewer 1 Report

The authors have done a good work in introducing the best DVR topology with reduce cost. The analysis of the performance of the DVR with PID and SMC controller under different fault conditions and also THD% of proposed controllers using FFT is very good. This work should be a good addition to the means of alleviating voltage sag.

Author Response

We are thankful to the reviewer for such as fruitful comments..

Reviewer 2 Report

This paper presents an operational comparison of a DVR topology based on an ac-ac three-phase matrix converter, when this device is controlled through PID and Sliding Mode Control (SMC) strategies, in order to cope with balanced and unbalanced voltage sags. Authors have utilized Ant Colony Optimization (ACO) algorithm to determine the controller gains, claiming that this strategy provides faster convergence and robustness to the controllers. Finally, through numerical simulations, it is shown that the sliding mode controller exhibits a faster response and without overshoot when compensating for voltage sags, while achieving load voltage signals with lower harmonic content.

This reviewer has several comments/suggestions/questions on the paper, which are listed below:

1.    It is necessary to improve the quality of figures, in particular figures 1 and 6.

2.    At the beginning of section 2.1 it is mentioned that the three-phase matrix converter is proposed in this manuscript, so it is suggested to improve the wording to indicate that only the device is utilized.

3.    In the title of section 2.2, ‘Par’ should be replaced by ‘Park’.

4.    Is section 3 really required as a separate section? This reviewer considers that this information could be omitted as it is basic concepts.

5.    On page 7 it is writing ‘The sliding surface is calculated through equation (14)’, but the statement refers to equation (13). Please review and correct.

6.    On page 8, it is mentioned that the rise time, settling time, overshoot and steady state mainly depend on the selection of alpha and beta values, but the equation (16) that defines the 3rd cost function only includes the beta variable. Authors are suggested to indicate to which beta variable they are referring to.

7.    Authors should indicate the values of the controller gains, calculated with the ACO algorithm, and utilized in the presented simulations.

8.    In figure 7, on the reference current label, replace ‘Refernce’ with ‘Reference’.

9.    Section 7 presents the analysis of the harmonic content of the load voltage signals. This reviewer considers it necessary for authors to indicate the moment of the simulation in which they performed the harmonic analysis presented in figure 12. Although authors point out that the THD% is an important factor to take into account for the application that is being analyzed, since IEEE Standard 519-1992 refers to this problem as a steady state phenomenon, the THD% value during the disturbance cannot be considered a parameter to evaluate the efficiency of the controller.

10.The paper needs proofreading. At many places, this reviewer is unable to understand what authors mean.

For more than 25 years the DVR has been adopted as a common solution for voltage disturbance compensation, and since then a wide variety of topologies have been proposed, including those using AC-AC converters in order to eliminate the drawbacks imposed by use of storage devices. While the DVR based on matrix converter is an interesting scheme to consider in terms of cost reduction, there are several important aspects that can affect the performance of the controller at the time the disturbance occurs, such as the source that will provide the energy for compensation, and which is not addressed in this manuscript at all. Even though the presented results show that the DVR controlled by the SMC strategy exhibits a very satisfactory response in terms of speed and overshoot during disturbances, such a paper should include a more detailed design of the controller, as well as stability analysis and validation through experimental results, in order to highlight the contribution of this proposal with respect to those already published in the literature.

Author Response

We are thankful to the reviewer for such as fruitful comments

Reviewer 3 Report

The research carried out by author is very good, the first time matrix converter has presented with sliding mode 

controller to get the better operational reliability of DVR with reduced cost, only minor corrections are needed in  this research paper. 

1. Figures are cleared but some figures are without X-Axis dimensions and units, so they must be incorporated. 

2. English grammar needs improvement and whole paper must be checked. 

10% of supply voltage is not acceptable and may cause failure of equipment, and revenue loss [2]. 

10% of the supply voltage is not acceptable and may cause the failure of equipment, and revenue loss [2]. 

The researchers were presented the different types of the FACTs device in shape of the SVC, STATCOM, Shunt capacitors for the voltage sag alleviation [3], but they are shunt connected devices. 

The researchers were presented the different types of the FACTs device in the shape of the SVC, STATCOM, and Shunt capacitors for the voltage sag alleviation [3], but they are shunt-connected devices. this only possible through application of series connected device [1]. The Dynamic Voltage Restorer 

(DVR) is connected in series with the load inorder to compensate the voltage sag of small and large value [6]. this is only possible through the application of a series connected device [1]. The Dynamic Voltage 

Restorer (DVR) is connected in series with the load in order to compensate for the voltage sag of small and large value [6]. 

The researchers were have presented the DVR topologies supported with batteries, capacitors for the supply of DC power to the inverter circuit of the DVR [7], but the drawbacks of this topology is are high cost of the maintenance and replacement [8]. In battery supported DVRs the multiple conversion of power supply is needed which increases the complexicityof the system [9]. 

The researchers were have presented the DVR topologies supported with batteries, and capacitors for the supply of DC power to the inverter circuit of the DVR [7], but the drawbacks of this these topologies is are the high cost of the maintenance and replacement [8]. In battery-supported DVRs the multiple conversion of power supply is needed which increases the complexicity complexity of the system [9].

Author Response

(The authors gave the same response as above.)

Round 2

Reviewer 2 Report

The manuscript has been enhanced and most of the proposed modifications have been addressed by authors.

In some of the figures the quality was improved, although in all the figures that contain the results presented in the revised version of the paper the quality deteriorated, which should be reviewed and corrected

In my early revision, an observation was made to the authors about experimentally validating the SMC strategy used in the DVR, in order to have strong evidence to support the effectiveness of the proposal. Authors argue that experimental setup is not feasible due to high cost of equipment related to this proposal, however, there is a large number of research works that present low-power prototypes in order to validate the proposed control strategies. Authors are suggested to consider this possibility for future proposals. Another observation made was that there is not mention of where the energy used for compensation is obtained, which I consider to be an important aspect that should be mentioned in the final version.

Author Response

We are much thankful to the reviewer for fruitful comments.
